# Analyzing a sport for development program's logic model by using key actors' perceptions: The case of Pour 3 Points organization in Montreal

**Tegwen Gadais** [1,2] *, **Mariann Bardocz-Bencsik** [3]

**1** Université du Québec à Montréal (UQÀM), Montréal, Canada, **2** UNESCO Chair in Curriculum Development (UCCD), Montréal, Canada, **3** University of Physical Education, Budapest, Hungary

* gadais.tegwen@uqam.ca

## Abstract

### Introduction

More work is needed on measuring the impact of Sport for Development (SFD) organization and on the managerial structures and processes for change. The purpose of the current study was to analyze the logic model (LM) of a SFD program in Canada that provides training for high school coaches in low socioeconomic communities in Montreal.

### Methods

Key actors (i.e., coaches, program administrators, school directors, and sport coordinators; N = 22) were interviewed about their perceptions of the different components of the organization's LM, namely the program's context, the initial problem it addressed, its needs, objectives, input, output, and impacts.

### Findings

Findings reveal the participants perceived the program as being successful by all key actors. Participants had similar understandings regarding the targeted problem and context, but their views differed regarding their understanding of the program's activities. In addition, the key actors addressed issues of the structure and impacts of the SFD program and made suggestions to improve the program, including clarifying its objectives, reinforcing internal communication, and building stronger partnerships with the partner schools.

### Conclusions

Findings from the present study provide recommendations to help improve the organization's LM. In addition, these findings can help researchers and SFD administrators reinforce essential organizational program structures and activities for better management, evaluation, and improved impact on communities.

**Data Availability Statement:** The Data contain potentially sensitive information and cannot be shared publicly because of Data Access / Ethics

Committee. Data requests can be sent to https://cerpe.uqam.ca for researchers who meet the criteria for access to confidential data.

**Funding:** The authors received no specific funding for this work.

**Competing interests:** The authors have declared that no competing interests exist.

# Introduction

This study contributes to the advancement of the field of research on sport for development (SFD). It is important to better understand the real impacts of these projects on the populations they address by producing scientific knowledge. The present study is the result of a collaborative research designed to improve the logic model (LM) of an SFD organization in Montreal called *Pour 3 Points*, by using various key actors perceptions.

## Sport for development (SFD) programs

Researchers investigating SFD programs have described benefits of sport participation, such as individual development, health promotion and disease prevention, promotion of gender equality, social integration, peace building or conflict prevention/resolution, and assistance post-disaster/trauma [1, 2]. Despite the potential benefits of sport, these positive impacts do not accrue automatically. Reaching positive impacts requires professional and socially responsible interventions adapted to the social and cultural context that give priority to developmental goals and are carefully designed to be inclusive [3–5]. Still, some authors note the lack of scientific literature concerning the understanding of the mechanisms by which sport can foster the development of participants [6–9]. In addition, researchers are calling for more work on the managerial structures and processes in SFD, by using creative inter-organizational collaborations between various partners among others [10]. On the other hand, SFD organizations are frequently asked for monitoring and evaluation studies to demonstrate accountability to funding partners, but their targets and strategies remain unclear and questionable to perfectly applied program evaluation protocols [11–14].

## Current limits on SFD project evaluation

A classical way to observe the impact of sport on social change is through SFD program evaluation [12, 15–17]. Evaluation studies have investigated various aspects on SFD projects' missions and paradigms [12, 18–20] but frequently failed at giving a clear answer on what are the impacts of the program. A literature review conducted by Levermore [12] found three major limitations to SFD evaluation studies: (a) monitoring and evaluation are insufficient; (b) they are conducted with acclaimed programs (e.g., meaning only well known programs have resources to be evaluated through a rigorous evaluation process); and (c) they have the tendency to employ a positivist logical framework. This literature focuses on SFD evaluation undertakes or not, methods used to assess the project, and public diffusion of the results. Current approaches have been criticized by participatory/critical viewpoints for their top/down and quantitative focus. Levermore concluded this literature review highlighting the need for evaluations that can address the diversity of SFD projects, some with unclear objectives or missing rationales. Programs need evaluation with strong methodological literature for logframe and critical participatory approaches on attempt to apply these approaches to selected case studies or consider their use in the context of a specific sports event [12]. Those limits have since been readdressed by more recent studies [21–23]. Coalter [11] proposes even a monitoring and evaluation tool specially adapted for SFD. However, resources on the SFD field remained frequently unprepared and not qualified for conducting a strong monitoring and evaluation process.

In addition, Lynch and Yerashotis [24] raised two essential questions to consider when evaluating SFD programs. First, is it relevant to investigate the methodologies used given the often dangerous and complex settings in which the research takes place? In particular, local settings of SFD projects are often unstable or insecure, and theoretical frameworks rarely address the contextual challenges of sport for social change practices [25–27]. The complexity

of most SFD contexts require the development of operative research method for direct data extraction [28, 29]. Second, should the research be used to support/reinforce the field practices or criticize and question actions and achievements [30]? Ridde and Dagenais [31] recommended engaging practitioners in a collaborative research, also identified by some SFD researchers to address contextual challenges [25–27, 32]. In this sense, the conceptual (theory) and operational (practice) understanding of the key actors (e.g., administrators, stakeholders, decision makers, funders, beneficiaries) within SFD projects, allowing researchers to better understand how SFD program operates on the ground, and to form recommendations for the organization to upgrade their own project on the field. In this sense, both perceptions of key actors involved in a program could be considered as valuable data, and be combined around a common language: the logic model of the program [33].

## Logic model

One way for improving program evaluation and achieving more tangible results is to reinforce the Logic Model (LM) of the project [31, 34, 35]. The Logic Model (LM), logical framework or log framework is a visual tool that gives individuals involved in a project a common understanding of the mission, vision and procedure through a conceptual map [36]. It is considered as the cornerstone of a project or a program and it allows the operational linkage of all the elements necessary for the implementation, management, and evaluation of a program. In SFD field, organisations and researchers also used the "Theory of change" as a process to illustrated goals, activity and impacts of a program. Both LM and Theory of change (ToC) could be considered similar or from the same family [37, 38]. In this study, researchers prefer to use the LM as referring to the international development standards and research field culture related to the program monitoring and evaluation [37–39] as well as the Coalter's work [11] and Commonwealth tools kit developed for measuring impacts of sport, physical education and physical activity [40]. Three main differences could be used to argue our choice. First, LM illustrates a programme through the understanding of the change process (implementation), while ToC gives a big picture to summarises work at the strategic level. Second, LM presents the intervention in a logical sequential way connecting components to outputs and concrete results. ToC gives more the complex social, economic, and institutional process that underlie societal change. And third, the LM answers the question if we plan to do X, then will give Y results, rather than questioning the mechanisms under the change.

LM's address three main common features present in program, namely *content* (what?), *stakeholders* and *beneficiaries* (who?), and *reason to be* (why?), which in turn are often divided into six categories: needs, objectives, program, activities, inputs, output, beneficiaries, results, and external factors and context [41]. Fig 1 represents a LM considering context and the program evaluation perspective. First, the initial or targeted problem (*needs*) emerge directly from the context where the project takes place, and is the major target of the project linked with beneficiaries' needs. Second, *objectives* address the problem and the intention of action through the project. Third, the *program* itself refers to the *activities*. The program means the intervention or actions that are implemented in the project to reach the objectives. The *activities* receive and constitute of *inputs* (e.g., funds, equipment, human resources, training) and *outputs* (e.g., products, activities, training, actions). Fourth, the program is directed to the *beneficiaries*, who are the targeted population of the program. Fifth, the *results* are represented with different times of effect (short, medium or long term). Finally, all projects are influenced by *external factors* (e.g., favorable or rough geography, good or bad politics) and the *context* of their implementation. All components of the LM have to be adequate and well articulated together to constitute the fundamental logic of a program [31, 34]. Porteous [41] reminds us

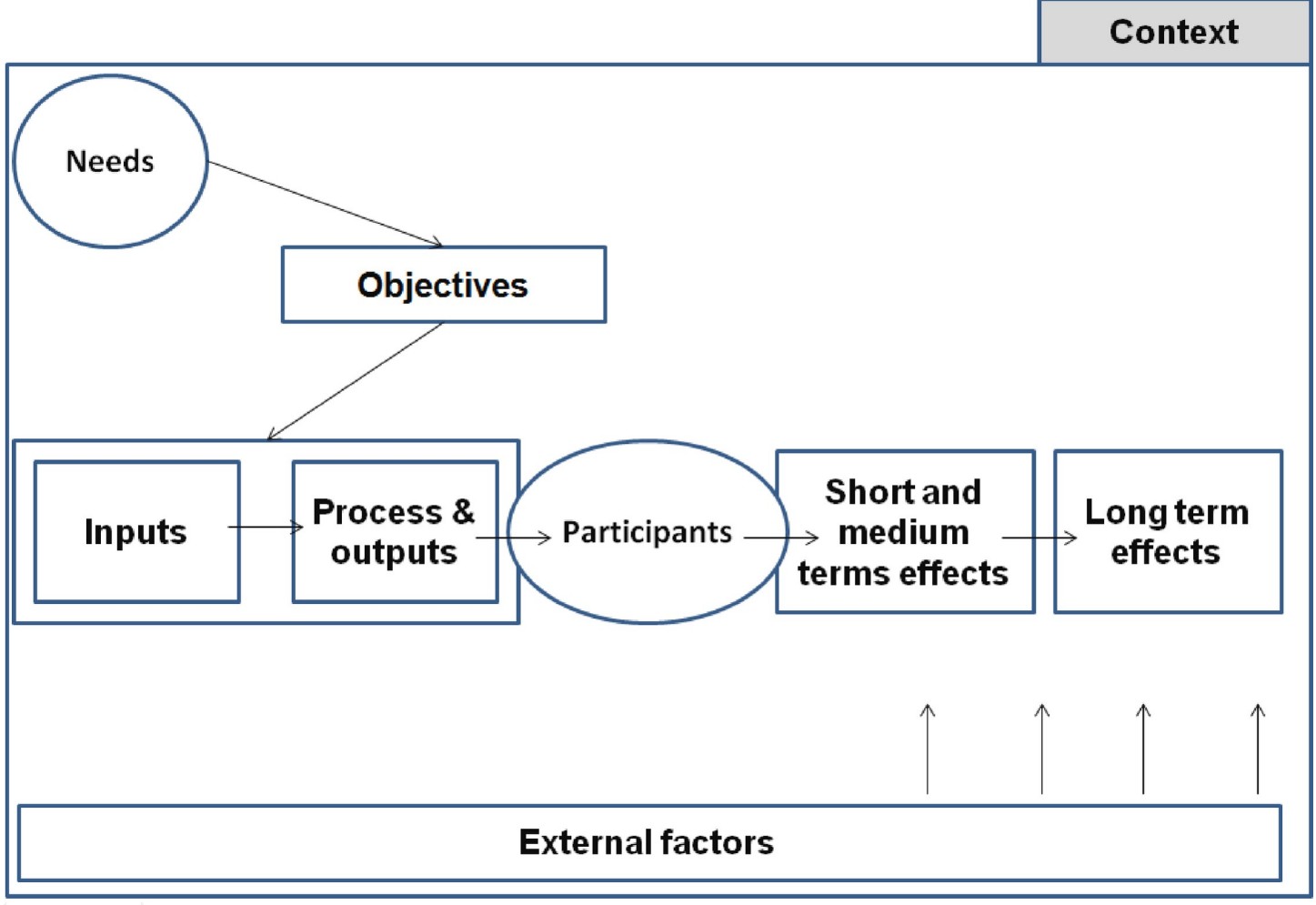

**Fig 1. Logic model representation and its components (inspired by Ridde & Dagenais, 2012).**

that it is essential to have a common vision of the project logic before thinking about monitoring and evaluation.

## Context of the study

This research was done in collaboration with a non-profit SFD organization Pour 3 points (P3P) established in 2013 in Montreal, Canada, that used sport as a tool to foster youth development in low socioeconomic neighborhoods. At the time of the study, the organization provided a two-year coaching training program (https://pour3points.ca/en/le-programme-de-certification-en-coaching-pour-3-points/) for young Canadian adults who were interested in coaching and were willing to take on the long-term engagement of the program. During their enrolment in training program, they obtained skills to become life coaches while coaching sports at one of the organization's partner schools. Each year, the program recruited approximately 15 coaches. Throughout their enrolment in the program, the coaches participated in a four-day training retreat, five peer discussion circles, five formal trainings sessions, and three personal evaluations each year (see S1 Appendix).

Before the start of the program, P3P administrators designed the first LM of the coaching program in collaboration with a consultant company. All of these administrators had a

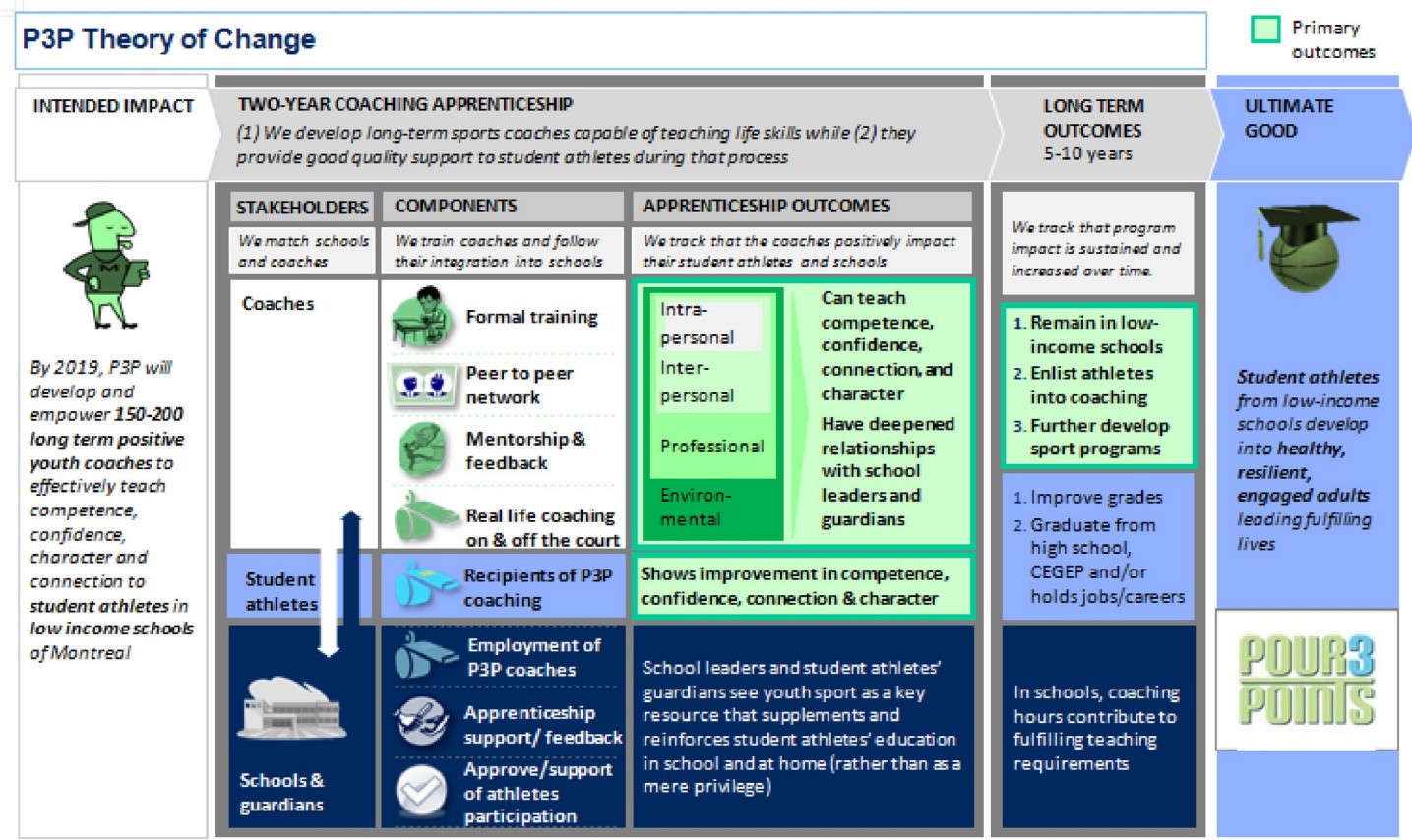

**Fig 2. First P3P training program's logic model.**

steering role in the organization such as running the organization or the coach training program. Together, they built the current LM (theoretical) (Fig 2) based on sport coaching literature, including the Positive Youth Development in Sport approach and its 4C's outcomes (Competence, Confidence, Connection and Character) [42, 43], as well as the types of coaching knowledge (i.e., intrapersonal, interpersonal, professional, and environmental) [44]. The administrators reached out to the research team with a request to assess the efficacy of their LM (theoretical and practical) as well as to suggest improvements.

## Aims of the study

The purpose of this study was to analyze the LM of a SFD organization in Canada (P3P) that provides training for high school coaches in low socioeconomic communities in Montreal. More specifically, authors aim to answer the following questions:

1. How do key actors (coaches, program administrators, school directors and assistants, sport coordinators) perceive an SFD program on the field (practice), compared to its LM (theory)?

2. What are the strengths and areas of improvement of the SFD program according the key actors perceptions?

3. Make recommendations to improve the LM of the organization and its components.

## Methods

### Realistic evaluation

The principles of *realistic evaluation* were employed as a gateway to collaborate with key actors [31, 33, 45]. This approach allows researchers to engage participants into the research process taking into account contextual challenges [25–27]. With regards to the present study, this approach enabled the research team to conceptualize the SFD program following the components of the LM (i.e., context, initial problem, objectives, activities, results and impacts) and then create direct link for program evaluation. Realistic evaluation considers the context and the expertise of the key actors to achieved program evaluation through building the LM from a down/top process [33].

### Participants

The participants of this study were 22 key actors (i.e., coaches, program administrators, school directors, and sport coordinators) involved in the program in Montreal (R = 18–55 years old). They were four administrators (3 men and 1 woman) responsible for program development, seven coaches (4 men and 3 women) who received the training and coached high school student-athletes, five sport coordinators (2 men and 3 women) in charge of the implementation of the school sport program and mentoring the coaches, four school assistant directors (1 man and 3 women) responsible for the school activities and programs, and two school directors (2 women). These participants were selected because of their involvement as decision-makers and their role in program implementation and also their availability and willingness to take part in the project and be interviewed.

### Data collection

Data were collected using semi-structured interviews between April and November 2017. Interviews were designed to examine the respondents' perceptions on different components of the program based on the seven components of the LM and to provide suggestions for improvements and to make note of program elements that functioned well. The participants also addressed aspects of the program that did not fit into these categories of the LM. The interviews were conducted by the first author, they took place in a calm and quiet place, were all audio recorded, and took on average approximately 60 minutes.

### Data analysis

Researchers conducted a combined strategy based on Braun and Clarke [46] and Yin [47], which followed seven steps: (a) transcription, (b) familiarization with the data, (c) coding, (d) identifying categories within each theme, (e) reviewing categories, (f) defining and naming themes, and (g) writing. The first stage of the data analysis consisted of transcribing the interviews verbatim. Second, two authors became familiar with the data by listening to the audio recordings and reviewing the transcriptions. Third, data was coded using a deductive (preestablish categories of the logic model) and an inductive process through the conceptual categories of Paillé & Mucchielli [48]. Fourth, lower-order themes were deductively identified within each overarching theme defined by the components of the LM. In the fifth and sixth steps, the first and second authors reviewed the overarching themes and lower-order themes, naming and defining each. Finally, the seventh stage consisted of disseminating the findings and telling the story from the perspective of the key actors, which is addressed in the results section. Authors organised the data into four categories: perceptions and patterns, strengths of the program, areas of improvement and then recommendations.

## Quality standards

Quality standards were applied to ensure quality on qualitative SFD research [49, 50]. Based on those authors, the following strategies were used: (a) width, (b) aesthetic merit, (c) worthy topic, (d) rich rigor, and (e) transparency. To achieve width (i.e., comprehensiveness and quality of evidence), the first author conducted interviews and collected data from all key actors, provided a detailed description of the data analysis with the help of the second author, and reported direct quotes of the participants to allow the reader to judge the quality of the data. Aesthetic merit (i.e., creative analytical practices) was addressed by using an inductive thematic analytical process, which opened up the text for explanatory interpretation of information. The study itself is deemed a worthy topic given that it originated from a request of the organization and was relevant, timely, and significant to their needs. The study showed rich rigor (i.e., use of theoretical constructs, abundant data, various P3P key actors) by using realistic evaluation and the LM as central theoretical frameworks and collecting 1320 minutes of interviews. Transparency was attained through regular discussions between the two authors from three various backgrounds in the understanding of SFD. In addition, bracketing was also used as a quality standard criterion. Bracketing promotes self-reflection and raises awareness to how one's personal experiences may impact the collection and interpretation of data [49, 51]. Bracketing is presented as two forms of researcher engagement: with data (identification and temporary setting aside of the researcher's assumptions) and with evolving findings (hermeneutic revisiting of data and of one's evolving comprehension of it in light of a revised understanding of any aspect of the topic. Both of these processes are ongoing, and they include the careful development of language with which to represent findings [52]. To meet this quality standard, documents (such as Theory of Change, Program description) and the first author's field notes (organised around principal categories explored in this study and the recommendations of Paré) [53], were used to better understand the context of the program, its needs, as well as the key actors' involved and their relationships. Complementary, TG has a profile of West Europa and North America background. He has been physical education teacher and humanitarian worker for 15 years, now researcher on SFD since almost 15 years, and MBB is from East Europa has been working in communication, as independent researcher and consultant around SFD for several years. These two profiles and backgrounds allow to challenge the perceptions of each author and avoid bias in the analysis. Finally, the study received ethical approval from the first author's host university (1087_e_2017). Participants (all majors) gave their writing consent to participate to this study by completing and signing a sheet form.

## Results

The results section is divided into three sub-sections: the first section addresses the responses given by key actors following the topics of the interviews, related to the elements of the LM, and available in Table 1: *initial problem targeted by the program*, *context of the program*, *objectives of the program*, *activities of the program*, *results*, and *impacts of the program*. Participants' quotes are used to illustrate the findings, and codes are included in parenthesis describing the role of the key actors in the organization, namely, administrators (A), coaches (C), sport coordinators (SC), school assistant directors and school directors (SD). The second section highlights strengths of the program while the third section is about the areas for improvement.

### Key Actors' perceptions and patterns

**Initial problem targeted by the program.** The wording "initial problem" has been used by the SFD organization in their communication, and it is also used in the literature instead of "needs", that is why the current study uses the same wording. Two main ideas were mentioned

**Table 1. Perceptions and patterns of the P3P program regarding the logic model and its components.**

| | Targeted problem | Context | | Program objectives | Activities | Results-Impacts | General Comments |
|---|---|---|---|---|---|---|---|
| **Coaches (C)** | School problems (4) (e.g., school dropout, academic failure) Disadvantaged children (3) | Children are unsure about what P3P is (1) The target group of coaches is important (1) Attractive program (2) | | Teaching values (2) Developing children (3) Impacting children positively (2) Developing life coaches (1) Motivating children to have goals (1) | Follow-up (4) The content of the educational activities needs to match the experience of the coaches (3) The timing of the activities is a concern (3) The amount of the activities is a concern (3) | Children not only seek victory, but learning as well (5) Coaches develop on a personal level (2) Children develop on a personal level (5) Children develop their basketball skills (1) The children develop perseverance (4) | |
| **P3P administrators (A)** | School problems (1) Disadvantaged children (2) | Children are unsure about what P3P is (1) The partnership/ relationship between P3P and the schools is challenging (3) | Teaching values (1) Helping children reach their potential (1) | The content of the educational activities needs to match the experience of the coaches (2) | The children develop perseverance (1) Children develop on a personal level (3) Coaches develop on a personal level (1) Question: Who are the targets of the impact? (2) | Comments about financing the work of the coaches (1) Status of coaches needs to be clarified (2) | |
| **School directors and assistants (SD)** | School problems (1) | The target group of coaches is important (2) Children are unsure about what P3P is (1) The partnership/ relationship between P3P and the schools is challenging (1) | Motivating children to have goals (1) Developing life coaches (1) | Information about the activities is missing (3) Ideas about homework help as an activity (2) | The behaviour of the children changes (5) Children develop perseverance (1) Children develop on a personal level (2) Coaches develop on a personal level (1) | Comments about financing the work of the coaches (2) | |
| **Sport coordinators (SC)** | Disadvantaged children (1) | The target group of coaches is important (1) | Helping children reach their potential (1) Developing life coaches (3) | | Children develop on a personal level (4) Coaches develop on a personal level (1) The children develop perseverance (1) Question: Whose impact are these? (1) | Comments about financing the work of the coaches (1) The sport coordinators and the coaches need to talk to each other (2) | |

by the participants when asked what they believed were the needs or the main problem targeted by the organization. *School problem* was mentioned by 6 respondents (4C, 1SD, 1SC) and it includes "lack of motivation", "drop-out", and "general problems at school". As a coach explained:

> "I grew up in this city, I attended a couple of high schools here. I saw the environment, I saw some coaches here, so I saw how much an organization like this is needed. They (the organization) has a concrete plan, they target environments where attending school is very difficult for young people".

*Disadvantaged children* (3C, 1A, 1SC) addressed "children living in disadvantaged areas" and "deprived neighborhoods", and "children with less chance". An administrator said that

*"Basically, we exist because the educational progress of young people in underprivileged environment is troubled, that is the starting point. That's the reason of our existence."* Taking together, the organization targeted children's school problems, as well as children's personal challenges of living in low socioeconomic environments.

**Context of the program.** With regards to the context of the program, some participants noted that the organization is very *attractive* from the outside (2C), people want to be part of it. Respondents talked also about that the *target group of coaches is important* (1C, 2SD, 1 SC), such as the coaches as a target group and their selection. At this point, participants mentioned the idea of having *"more than one coach per school"* and the need *"to specify the target group of coaches"* (e.g., experience, type) as suggestions for improvement. It was also mentioned by a school director that a positive aspect of the selection process was that coaches were old students of the schools, who *"already have a belonging to the school, and can make the link with current students"*. It was also said that, *"the activities were successful, because they have old students as coaches who already know the philosophy"*. When talking about the children's knowledge on the program, related to the impression that *children being unsure about what P3P is* (1C, 1A, 1SD), it was mentioned that *"the children don't know the organization and only coach take part in the training"*. It was also stated as a suggestion for improvement that the children need to *"get to know the program"*. Communication is missing for children to understand the program, its goals and activities. Following this point, participants noted a lack of information coming from the SFD organization to the schools and also that the *partnership or the relationship between P3P and the schools is challenging* (3A, 1SD), or unclear and needs to be clarified.

**Objectives of the program.** The participants described the objectives of the program as *teaching values* (2C, 1A), *developing children* (3C), *having a positive impact on children* (2C), *developing life coaches* (1C, 1SD, 3SC), *motivating children to have goals* (1C, 1SC) and *helping children reach their potential* (1A, 1SC). As a coach said *"The objective is having a positive impact, not only an immediate one, but a long-term impact on the players. An impact that could make for example a student finish high school, who otherwise wouldn't finish it."* From the six objectives that were mentioned only one has the coaches as target group, namely "developing life coaches"–also phrased as "transforming coaches to life coaches" and "preparing/training life coaches". As a school coordinator explained:

*"The objective, I think, is to guide the coach to be able to manage all kinds of situations with the young people, to give them the most possible tools to become positive leaders in the community, then, just to encourage them to use this model in practice."*

Some respondents meant "teaching/transferring values to children"–through sport–when talking about teaching values, while others mentioned "teaching values" or "teaching SFD program values", not specifying the target audience.

**Activities.** This overarching theme addressed multiple program interventions and activities (what they are and how they are perceived). More specifically, the *follow-up* activity of the coaches (mentors meeting the coaches daily) was mentioned (4C) as one of the activities of the program. As a coach said: *"The activities are quite varied. We have our one-on-one follow-ups with our mentors, then we have our discussion circles with our peers. I could even take more of them, I'm very engaged in them, I love them."* It was also stated that the *content of the educational activities needs to match the experience of the coaches* (3C, 2A). An administrator pointed out that *"year #1 and #2 need to be separated at formations"*, while a coach said that *"the formations need to be adjusted to the experience of the coaches"*. The *timing of the activities* was mentioned as a concern (3C), stating that *"it would be better to have the formations before the season"* and that *"the rhythm of the activities needs to be adjusted"*. There were comments also on the *amount of the activities*

(3C), suggesting some additional activities on "extreme situations", "addiction", "attention deficit hyperactivity disorder (ADHD)" and "techniques and strategy". Another coach gave an example of what sort of additional training would be useful for them:

> *"I would love to have a training on ADHD (attention deficit hyperactivity disorder) and other behaviour-related problems, because it is said that the majority of the students have these problems. It would be good to know about them. Of course, I can do my own research, but having a lecture on them would be really good."*

It was also pointed out that the *information-sharing about the activities is missing* (3SD), as some respondents said that they *"don't know about them"*. The idea of *homework help* was mentioned twice (2SD). Someone said–referring to a school—that *"all sports get the obligatory homework help, it is between school and trainings"*, while someone raised the question whether *"the program is able to build a concept where the coaches help with the homeworks"*.

**Results and impacts.** The interview had separate questions about the perceived short-term results and longer term impacts of the program, also distinctly focused on coaches and on children, yet the respondents did not make this distinction at all times. Therefore a decision has been made to treat these two topics as one. The two words "result" and "impact" is used interchangeably throughout this section. The following items were mentioned when answering the questions about results and impacts: the *children not only seek victory, but learning as well* (5C), the *children develop on a personal level* (5C, 3A, 2SD, 4SC)–i.e. in "decision-making", "building trust", "developing autonomy", "pride" and "sense of belonging"—the *children develop perseverance* (4C, 1A, 1SD, 1SC), the *coaches develop on a personal level* (2C, 3A), the *behaviour of the children changes* (5SC)–for example "they are not late", and the *children develop their basketball skills* (1C). As an administrator said: *"The core idea of this program is to develop coaches, so that they can intervene and support the educational progress of their players. So, the focus is really on the development of the coaches."* A coach completed:

> *"I would like to see that school perseverance develops in the kids. Some of my players had and still have difficulties at school, and I wish I can be a part of their stay in school. On Mondays and Wednesdays, I spend some time with them, I help them with their homework, and it warms my heart."*

But at the end, a question remains *who are the targets of the impacts* (2A, 1SC), coaches or students?

**Other aspects of the program.** When given the opportunity to add other comments on the program, the respondents had diverse thoughts. There were comments about *financing the work of the coaches* (1A, 2SC, 1SC). A respondent suggested that *"with payment, the coaches might perform better"*, and others said that they *"give the coaches a small remuneration"*. Another respondent said that it is *"good to have the P3P coaches for free"*. It was also mentioned that the *status of coaches needed to be clarified* (2A), referring to the contractual relationship that the coaches have with P3P. Someone pointed out that *"the coaches have a contract, but if they don't show up at trainings, there are no consequences"*, while someone else posed the question whether *"the coaches are employees of P3P"*. The *partnership/relationship between the SFD organization and the schools* was also commented on (3A, 1SD), stating it as "challenging" and "unclear". As one of them explained:

> *"The approach to the partnership between the organization and the schools is not clear. Do we develop something with the schools and then work on it in the future? Do we need to partner*

*with the schools at all? For the schools, it is very confusing that the coaches are in training, but on the other hand, they look like our employees. We need to define their role, because being in training is not the same as being employed.”*

A school leader said:

*“We have excellent coaches, they are very engaged with our school. Nonetheless, their immediate superiors are the sport coordinators, and we need to improve the communication between them, so the parents could be more informed as well.”*

It was also mentioned that *“the organization lacks visibility internally in the school, therefore, they need to promote their partnership”*. Similarly, someone else said that *“the communication can be improved, as the school leadership doesn’t see what P3P does.”* As a general comment it was also pointed out that *the sport coordinators–paid staff members of the schools who are responsible for school sport programs, including the one of the organization—and the coaches need to talk to each other* (1A, 1SC), but according to a sport coordinator *“there’s no time for it.”*

## Strengths of P3P program

The data analysis also tried to highlight the positive aspects of the program pointed out by the respondents (Table 2). Globally on the needs, all respondents agreed that the *initial idea of the*

**Table 2. Strengths, areas of improvement and recommendations about the Logic model of P3P program.**

|  | Needs—Initial problem | Context | Objectives | Activities | Results and impacts | General comments |
|---|---|---|---|---|---|---|
| **Strengths** | **Initial idea of the P3P program is relevant** training and support for coaches (focus on coaches) qualified coaches inside school (focus on kids' development) | P3P advertisement and visibility is **attractive** |  |  | **Program affects coaches on personal and professional** level Program helps to change the behaviour of certain kids Positive changes with good potential to be sustained in the long-term |  |
| **Areas of improvements** | Discrepancy within **the logic model around the intended impact** of the program. It states that the intended impact is to develop positive youth coaches, but the need for this education is not apparent in the LM, neither it is perceived by the respondents | **Lack of purposeful communication** to the children about the program. **Partnership/ relationship between P3P and the schools is unclear**, therefore, it needs to be clarified and there is a lack of information coming from P3P to the schools | **Misalignment between the program objective** regarding the coaches in the LM and the perception of the P3P staff members | **Divide between the coaches and P3P staff members versus the school personnel** in defining activities (Activities provided to the coaches, activities for kids) | **Four outcomes** in the LM are not mentioned a) coaches having a deepened relationship with school leaders and guardians, b) remaining in low income schools, c) enlisting athletes into coaching and d) further developing sport programs Basketball skill development is not in the LM | **Financing and status of the coaches** is a concern for key actors, therefore, it needs to be clarified |
| **Recommendations** | Need for life coaches should be included in the LM | Program shall be introduced to the players at the start of the season The partnership between P3P and the schools needs to be clarified | P3P staff members need to be clear on whether developing life coaches is a program objective or not | **Clarify and communicate** to all key actors what are the program activities and their content | Some **measures have to be taken** to close the gap between the perceived program results and the ones in the LM | **Status of the coaches** needs to be clear for all key actors |

*program is relevant* and personally they are positive about taking part in it for two reasons. On one hand, they agreed with the idea of providing training for coach, because they generally miss coaching experience and support in their practice (Point of view of P3P administrators and coaches). On the other hand, school staff members are happy to receive qualified and trained coaches to work with their youth population, for low budget and voluntarily (Point of view of sport coordinators and school directors). This tends to reveal that motivation and targeted problems are different in between administrators and coaches from one side, and school staff from the other side. They formed their opinions on the situation and activity regarding both the kids and the coaches, and it means that the respondents consider both of these groups to be beneficiaries of the program.

Then on the context, *P3P advertisement and visibility seemed attractive* because three coaches mentioned that they wanted to be part of it from outside. It seems to be a great opportunity, they are motivated to be part of the P3P community, and the program is "cool", in their words.

Regarding results and impacts, several coach respondents gave lengthy practical examples on how the *program affected them on a personal and professional level*, and they also gave us concrete examples on how the *behaviour of certain kids changed* over the course of the program in certain situations. Finally, the scope of this study is not to showcase these individual examples, it is worth mentioning that some *great positive changes* have already occurred to some program participants and these changes have a good potential to be sustained in the long-term.

## Areas for improvement

Table 2 presented also seven areas of improvement identified in this study. First, there is an *apparent discrepancy within the LM around the intended impact of the program*. The LM states that the intended impact is to develop positive youth coaches, but the need for this education is not apparent in the LM, neither the respondents perceive it. The targeted problems perceived by the key actors somewhat resonate with elements in the LM. The LM does refer to "student athletes in low income schools", however, it does not mention school problems as a targeted problem. The LM also mentions the coaches' ability–developed through the program—to teach "competence, confidence, character and connection" and the development and empowerment of "positive youth coaches", however, none of the perceived targeted problems is about the coaches.

Second, there is a *lack of purposeful communication to the children about the program*. The LM does not indicate how the coaches are being selected for the program, however, it is stated that "we match schools and coaches". The LM also states that one of the program components it to "follow the coaches' integration into the schools", but it does not refer to any explanation to the children about the program itself.

Third, there is a *misalignment between the objective in the LM and the perception of the P3P staff members*. According to the LM, the program's intended impact is (by 2019) to "develop and empower 150–200 long term positive youth coaches to effectively teach competence, confidence, character and connection to student athletes". With this objective, P3P targets the coaches to transfer values to children. There is an apparent gap in the responses of the P3P staff members, as nobody in this category mentioned the development of life coaches as a program objective, and this is the only category where it was not mentioned. There are *five more objectives perceived by the respondents*, most of them targeting the children: teaching values (not specifying the target group), developing children, having a positive impact on children, motivating children to have goals and helping children reach their potential (all four of them having the children as target group).

Fourth, the respondents had varying knowledge on and understanding about the activities. A pattern can be seen that there is a *divide between the coaches and P3P staff members versus the school personnel in defining activities*. The first group referred to the activities that P3P provides to the coaches, while the second one talked about the missing information about the activities and mentioned the extracurricular homework help as activity. According to the LM the program components are "formal training", "peer to peer network", "mentorship and feedback"–referred to by the respondents as "follow-up"—and "real life coaching on and off the court". The latter one is not perceived as a program activity by the respondents as no answer mentioned that the coaches carry out activities with the children in sport (i.e. trainings, competitions). There is also a *lack of information-sharing about the activities* among the coaches, the P3P staff members and the school personnel.

Fifth, *in the LM there are four outcomes that are not mentioned by the respondents* as program results: the coaches having a deepened relationship with school leaders and guardians, remaining in low income schools, enlisting athletes into coaching and further developing sport programs. Based on the LM, the program's intended impact is to "develop and empower 150–200 long term positive youth coaches to effectively teach competence, confidence, character and connection to student athletes". This resonates with the perceived impact on the coaches' development on a personal level. The LM lists apprenticeship outcomes and long-term outcomes as well. As apprenticeship outcomes, the coaches' ability "to teach competence, confidence, character and connection to student athletes" and to have "a deepened relationship with school leaders and guardians" are listed in the LM. The latter one was not mentioned as a result of the program by any respondent. In the LM as long term outcomes the following items are listed: "remain in low income schools", "enlist athletes into coaching", "further develop sport programs", "improve grades" and "graduate from high school/CEGEP and/or hold jobs/careers." From this list, the latter two are perceived by the respondents, seeing the children' perseverance as a result of the program, however the first three perceived outcomes are not mentioned as results of the program. Nonetheless, *the development of the children' basketball skills is a perceived result that is not listed in the LM*.

Sixth, among their general comments the respondents mentioned the unclear *status of coaches and the financial aspects of their work with P3P*. In the LM there is no mention of either of these aspects. Finally, the partnership/relationship between P3P and the schools is perceived to be unclear, and there is a lack of information coming from P3P to the schools about the program in general and about its activities in particular.

## Discussion

At this point, it is important to recall this study emerged directly from the administrators and their need to reinforce the LM before proceeding to program evaluation asked by their funders. Throughout the principles of realistic evaluation [31] and components of the LM, researchers go step by step with practitioners (key actors) to produced recommendations for the future of the SFD organization and its training program. Authors also provided lessons learned about using the LM as a tool to prepare the program monitoring evaluation process with SFD organisations.

### Recommendations for P3P

Table 2 presents a summary of recommendations made by the participants to improve the program and its organization. This report has been provided to the administrators in order to help them upgrade the LM of the program. The recommendations were summarized in three points.

First, the *organization provided three distinct services/mission* that were not clear to all key actors, namely life coach training program, coaches who work with disadvantaged youth in schools, and the mission of the P3P organization. Findings of this study addressed a question around the target group of the program: is it for coaches or for children? This misunderstanding is also expressed in the three objectives of the program (Fig 2): This organization is a training program for coaches, its develops children through sport, and its social mission in relation with schools and more largely the society. Authors recommend that the P3P staff members need to clarify the objective of the program rather than targeted participants, in particular whether or not developing life coaches is a program objective.

Second, the *SFD organization has to provide better communication* with coaches, schools, partners, and children. The organization seems to have identified the lack of training for youth educators as a challenge in the field but the general need for life coaches is missing in the LM. Consequently, it is important to clarify and communicate to all key actors what the program activities and their content are. It also needs to be decided by P3P whether life coaching on and off the court is a program activity. Then, communication with partners needs to be improved about the mission. The program could be introduced to the players at the start of the season. Therefore, the children would have an understanding of the purpose of the program and the key actors around them. Furthermore, some measures are needed to close the gap between the perceived program results and the ones in the LM.

Third, the *partnership between P3P and the schools needs to be clarified*, the contract between them need to list the rights and responsibilities of both parties, along with consequences of non-fulfillment. The contract needs to state the information-sharing obligation of P3P towards the schools. Therefore status of the coaches needs to be clear for all key actors, for example by clarifying their rights and responsibilities in their contract. The SFD organization and the schools need to discuss their financial strategy about the remuneration of coaches.

Fourth and based on previous studies, after identifying the strengths and areas of improvement of the training program, the next step as future recommendation would be to *co-build or reinforce the LM with P3P administrators to precise how sport can support development* [7, 54, 55] and how the P3P program will be evaluated [11, 12] through a robust LM and details indicators [37–39]. The organization chose to publish the results of this study to share the importance of research collaboration on long term perspectives, even if their current training program has already reached another step of development [12]. Still, working closely with a research team is a benefit, despite work tasks, agenda, methods, are grounded in two different realities with two timelines. Research teams would like to celebrate P3P for their willingness to engage with independent researchers to explore their trainings and interventions. At the end, researchers hope that by providing a good practice example for SFD organizations of self-reflection give them the opportunity to improve their project with or without the help of academic researchers.

## Lessons learned

Following the recommendations of Ridde and Dagenais [31], realistic evaluation was used for bridging theory to practice and generate data in order to address limitations on monitoring and evaluation in SFD. Several elements have been useful to reach this agenda and identify knowledge on context, effects and mechanisms of the intervention [56]. In particular, practitioners' perceptions are considered as valuable data because of the expertise and experience they have around the program and the context of implementation. In this study, various key actors around the SFD organization have diverse and complementary perceptions (training and activities perceived; different understanding of communication), especially about P3P

thought program (in theory) and P3P in action program (in practice). But all can help the research team to better understand what really the P3P program is about and reinforce ultimately the LM. Few previous case studies on SFD used various points of view to generate a strong understanding of phenomena [57–59] even if those studies are rich in terms of understanding the situation and the context and preparing the full assessment of a program. Furthermore, realistic evaluation allows us to support and strengthen the work of key actors around the SFD program while continuing to critically question their actions and their achievements [24, 31, 60]. By analyzing their perceptions and opinions, recommendations for key actors could be made to enhance the LM of the partner organization. It helps the organization provide better service within the program. From now on, it is up to the organization to develop robust indicators in line with their theory of change in order to prepare for the evaluation of their program and demonstrate the impacts to donors. According to author's knowledge, current research on SFD settings is often critical and rarely addresses the contextual challenges of sport for social change practices [25–27]. Finally, realistic evaluation always attempts to build the rationale behind the project, through intervention/intermediate/general theory and an iterative process [56]. It could be considered as a method to prepare monitoring and evaluation of the program by having clear targeted participants, objectives, activities among others [12, 31, 41]. This method of rebuilding and strengthening the LM allows generating effective indicators for future program assessment despite the limited resources and capacities of the organization. It also ensures that the right elements of the program are evaluated.

Another lesson learned from this case study is related to the LM as an effective tool to monitor and evaluated programs, recalling the work of Coalter [11], the propositions of the Commonwealth [40] as well as international and cooperation studies [34, 37, 38]. Authors recommend the organisation to develop a strong and robust LM while initiating their interventions or workshops guided by populations needs through their ToC. Authors believe both LM and ToC are necessary to deliver an effective intervention to participants as well as evaluate the perceived impacts of the intervention. The Theory of Change do not provide the same level of precision and details the LM and its indicators or categories can offer for preparing the evaluation of the program. This case study showed the ToC did not allow for accurate measurement of the P3P program's impact. Complementary, it could be pertinent consider using *Theory of the Program* to have a more macro perspective [61] or *Theory of Action* to focus more on the real mechanisms of the intervention [62]. Let recalls that SFD organisations and staffs remained frequently unprepared and not qualified for conducting a strong monitoring and evaluation process. LM can be fit for a purpose but if it is misunderstood or incorrectly implemented it became useless in practice. Those remains frequently nobles' intentions because SFD staff are not qualified or don't have time to conduct the program evaluation process. In this sense, it could be interesting for funders to support better the organisation with this heavy process.

In addition, LM is an instrument developed in international development to design the program theory (namely content, stakeholders and reason to be). If it is well designed and implemented, researchers can have a clear idea about why and how the program is working. It helps to prepare program evaluation and measure the perceived impact of an intervention. This element is common for both the realm of sport and the realm of social intervention (e.g., social work, community work or developmental work). Because both realms have the same indicators and SFD could be consider as an extension using both realms. They address needs of populations, target objectives, implement activities to reach impacts. More than that, LM makes the link in-between both realm, connecting social intervention and sport by explain how they are connected.

Finally, this study was intended to be a model of good practice and development between research and practice on SFD in order to use the knowledge and skills of both partners. In

addition to the use of the principles of realistic evaluation and LM, learnings can be retained for SFD actors. SFD practitioners should remember from this study that they need to think through their SFD projects by proposing and developing a clear, adapted and precise LM that effectively illustrates the interventions they are carrying out and the impacts they are aiming for. This clarification work, which must be done by professional's efficiency trained in program evaluation, will make it possible to establish the needs, objectives, activities and results to be achieved by the SFD programs. Obviously, the whole process will be supported by robust indicators to document the process and its results. In this sense, funders could also help train or provide resources to organizations. As for the researchers, they must propose support solutions to SFD organizations in order to help them achieve their objectives while minimizing the time and energy required for an already busy field organization structure. Several tools are already available for bridging theory to practice in the SFD field, such as the Actantial Model [28, 29], Coalter's manual [11], toolkit and associates indicators of the Commonwealth [40] or other collaborative program evaluation approaches [63] (Desgagnés) or co-construction process [55, 64].

## Limitations

First, it would be relevant to collect data from actors such as the young athletes, their parents and funders to share their opinion about the P3P program. Their opinion could provide additions to see the entire picture of the P3P program, and could be the basis of recommendations for further improvement. Second, some of the interview questions were not relevant to some respondents (e.g., school directors), as they didn't know much about the P3P program as such, therefore, other questions need to be considered in future research. Third, the researchers are aware of that this study has taken place in Canada, and conducting the same type of study in developing countries, could be more challenging for research. Nonetheless, the authors have diverse backgrounds (North America and Europe) and experience in international development and SFD projects, that provide a global look into the interpretation of the findings. Finally, the next step in this work would have been to assist the organization in developing the indicators needed to conduct the program evaluation. Unfortunately, for reasons mentioned in the introduction (redefining the context and reality of the organization), this step was not carried out directly following this research.

## Conclusion

This study was the result of a collaborative research for improving the LM of a SFD organization in Montreal: P3P. By using key actors perceptions throughout a realistic evaluation, this study contributes to the advancement of knowledge on SFD program evaluation. It generated strengths, areas of improvements and recommendations for P3P program such as clarifying its objectives, reinforcing internal communication, and building stronger partnerships with the partner schools. Findings reveal also that the program is perceived as successful for all key actors and help researchers and P3P administrators reinforce essential organizational program structures and activities for better management, evaluation, and improved impact on communities. The study also addressed some gaps and challenges for practitioners and researchers by using the LM. This study shows that the co-construction of a strong LM based on the perceptions of the program's actors, its context and its reality prove to be relevant in helping the organization to properly evaluate its work.

Others various findings emerged from this study: 1) Participants have valuable perceptions that help us better understand what the SFD program is really about, especially from perspectives of different key actors; 2) Collaborative research is useful to support SFD program by

understanding actors and their context; 3) Building a strong LM is essential before assessing a SFD program with clear indicators and measures, with someone qualify. Since the time of this study, the organization is aware of those results and currently applying improvements. This research is only the first through a long-term collaboration that the researchers would like to establish with this SFD organization.

## Supporting information

**S1 Appendix.**
(DOCX)

## Acknowledgments

Authors would like to acknowledge the P3P organization (administrators and coaches) and its partners (sport coordinators and school administrators) who took part to this study.

## Author Contributions

**Conceptualization:** Tegwen Gadais.

**Data curation:** Tegwen Gadais, Mariann Bardocz-Bencsik.

**Formal analysis:** Tegwen Gadais, Mariann Bardocz-Bencsik.

**Methodology:** Tegwen Gadais.

**Project administration:** Tegwen Gadais.

**Validation:** Tegwen Gadais.

**Writing – original draft:** Tegwen Gadais.

**Writing – review & editing:** Mariann Bardocz-Bencsik.

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
