## [Decision Letter · Decision Letter 0]

5 Oct 2021

PONE-D-21-03672Analyzing a SDP program’s logic model with key actors’ perceptions The case of Pour 3 Points organization in Montreal.PLOS ONE

Dear Dr. Gadais,

Thank you for submitting your manuscript to PLOS ONE. After careful consideration, we feel that it has merit but does not fully meet PLOS ONE’s publication criteria as it currently stands. Therefore, we invite you to submit a revised version of the manuscript that addresses the points raised during the review process.

Please consider all comments

We look forward to receiving your revised manuscript.

Kind regards,

Ahmed Mancy Mosa, Ph.D.

Academic Editor

PLOS ONE

Journal Requirements:

2. Please include your tables as part of your main manuscript and remove the individual files. Please note that supplementary tables (should remain/ be uploaded) as separate "supporting information" files

4. Thank you for stating the following financial disclosure: "The funders had no role in study design, data collection and analysis, decision to publish, or preparation of the manuscript." 

5. Please ensure that you include a title page within your main document. You should list all authors and all affiliations as per our author instructions and clearly indicate the corresponding author.

Reviewers' comments:

Reviewer's Responses to Questions

**Comments to the Author**

1. Is the manuscript technically sound, and do the data support the conclusions?

Reviewer #1: Partly

Reviewer #2: Yes

2. Has the statistical analysis been performed appropriately and rigorously? 

Reviewer #1: N/A

Reviewer #2: N/A

3. Have the authors made all data underlying the findings in their manuscript fully available?

Reviewer #1: Yes

Reviewer #2: No

4. Is the manuscript presented in an intelligible fashion and written in standard English?

Reviewer #1: Yes

Reviewer #2: Yes

5. Review Comments to the Author

Reviewer #1: Firstly, congratulations to the authors on a worthy piece of research focussing on often overlooked process of evaluation with SfD. See below for some specific and general comments.

Line 63 – Please clarify what you mean by acclaimed programmes and why this is negative. This term on its own does not clearly portray the issue within the cited paper. (11)

Line 116 – More detail required on what P3P actually delivers and what their implementation looks like in terms of the types of training they deliver, how this training is used within schools, does it have a timescale, is it only a training programme or does it provide more structure within the schools? I was still unclear by end of paper what the P3P model looked like, and it is discussed in very general terms. Example trainings or activities would be beneficial (can be included within Appendix if word count is an issue).

Line 181 – What kind of coding utilised?

Line -209 – Touched upon here but would like to see more detail around reflexive practices carried out among the research team. Giving an example of bracketing process used may help.

Line 214 – More detail around field notes, what they consisted of and their purpose within rigour of study would be beneficial.

Line 228 – Personally I would like the theme school problems to be a bit more descriptive, when I hear the term I think of problems with the school structurally whereas the description breakdown looks more at personal challenges faced by young people at school.

Line 465 – o to or typo

General Comments

The aims of the study focus on assessment of a logic model however there are many parts of discussion (and abstract) lean into impact. The study is not designed to explore effectiveness of associated impact, nor would this be an appropriate study design to do so, however some claims ‘Positive changes with good potential to be sustained in the longterm’ do make such claims. Please review language around such comments noting these are perceptions of involved stakeholders. Later mention of need to continue work with participants, parents etc does acknowledge this but review language to avoid confusion.

Secondly, I agree with your recommendations around the need for clarification around multiple aspects of the logic model and its implementation. A logic model can be fit for purpose but if it is misunderstood or incorrectly implemented it is of little practical use. There could be stronger language around how ill-defined key components of the programme appeared to be for participants within your data. For example, the discrepancy as to what the intended impact of the programme is constitutes a huge problem. As you highlight in the introduction the development outcomes need to be prioritised within SfD and a lack of clarity around them is a huge flaw within programming. I understand the need for sensitivity with partner organisations, but this is the purpose of this study and without such clarity future aims around effectiveness evaluation cannot occur. These recommendations exist within Table 2 but the importance/strength of these recommendations is not always mirrored in the discussion. These can be strengthened while still celebrating P3P for their willingness to engage with independent researchers to explore their intervention.

Reviewer #2: This is a well written and organised paper, with a coherent narrative and argument, and which moves towards enhancing our knowledge of the use of logic models in the Sport for Development field.

I have only a few suggested issues for the authors to consider:

- There could be a more rigorous discussion of the role of logic models in both the realm of sport (not just SfD) and the realm of social intervention (social work, community work, development work). SfD sits at the intersection of these two realms, so there needs to be some consideration of where logic models fit in, in these respective fields.

- Some justification of the focus on logic models compared to ‘theories of change’ should be given. The latter tend to be more prominent in SfD agencies and programs, so why focus on the former here?

- I’m surprised that there is not more discussion of Coalter’s work with respect to the critical evaluation of SfD programs. Coalter’s recent work on program theory may be particularly salient here.

- The paper could and should respond more adequately in the conclusion to the routine ‘so what?’ question. There should be greater consideration of what these findings may mean for future research and studies, as well as practice, in the SfD field, and also, perhaps more importantly, for researchers in broader fields in sport or community/social development or youth work. These latter points are especially relevant given the audience of the journal, which is far wider than the sports studies field.

6. PLOS authors have the option to publish the peer review history of their article (what does this mean?). If published, this will include your full peer review and any attached files.

Reviewer #1: No

Reviewer #2: No

---

## [Author Response · Author response to Decision Letter 0]

7 Feb 2022

REVIEWER 1

COMMENTS RESPONSES TO REVIEWERS

Reviewer #1: Firstly, congratulations to the authors on a worthy piece of research focussing on often overlooked process of evaluation with SfD. See below for some specific and general comments. 

Thanks for your kind comments. 

Line 63 – Please clarify what you mean by acclaimed programmes and why this is negative. This term on its own does not clearly portray the issue within the cited paper. (11) Please note that Levermore stated these 3 limits in his review of the literature concerning SDP, we just referenced their study. The author referred to the idea that only well know programs of SDP are evaluated through a rigorous process of evaluation because organisation have resources to do so. 

We now made a few modifications in the text concerning this point (l.64-65).

Line 116 – More detail required on what P3P actually delivers and what their implementation looks like in terms of the types of training they deliver, how this training is used within schools, does it have a timescale, is it only a training programme or does it provide more structure within the schools? I was still unclear by end of paper what the P3P model looked like, and it is discussed in very general terms. Example trainings or activities would be beneficial (can be included within Appendix if word count is an issue). 

Details of the program are mentioned on page l.140-143 (Each year, the program recruited approximately 15 coaches. Throughout their enrolment in the program, the coaches participated in a four-day training retreat, five peer discussion circles, five formal trainings sessions, and three personal evaluations each year). We added Appendix A and the organisation’s website site, which describes the present program and its schedule.

Line 181 – What kind of coding utilised? 

Coding was conducted following deductive (preestablish with elements of the logical model) and inductive processes based on Paillé & Mucchielli (2021) conceptual categories. After reading the verbatim, content was placed into groups of ideas and subgroups. We made modifications in the text concerning this point L.234-239.

Line -209 – Touched upon here but would like to see more detail around reflexive practices carried out among the research team. Giving an example of bracketing process used may help. 

This comment is relevant, and we now provide details on this aspect. Modifications were made in the text (l.227-231). We hope that our response address’ your comment.

Line 214 – More detail around field notes, what they consisted of and their purpose within rigour of study would be beneficial. 

We added some details and a reference about the field notes, we hope the rigour is better now. (l.232)

Line 228 – Personally I would like the theme school problems to be a bit more descriptive, when I hear the term I think of problems with the school structurally whereas the description breakdown looks more at personal challenges faced by young people at school. 

Thanks for this comment. We now give details about it (see table 1). 

Line 465 – o to or typo 

We made the change.

General Comments

The aims of the study focus on assessment of a logic model however there are many parts of discussion (and abstract) lean into impact. The study is not designed to explore effectiveness of associated impact, nor would this be an appropriate study design to do so, however some claims ‘Positive changes with good potential to be sustained in the longterm’ do make such claims. Please review language around such comments noting these are perceptions of involved stakeholders. Later mention of need to continue work with participants, parents etc does acknowledge this but review language to avoid confusion. 

Thanks for your comments. You perfectly understand what the aim of this study is and what we intend to do.

Following your comments, we decided to elaborate more on the impact of our study in the discussion and the abstract. 

Also, we change some inappropriate claims along the text about exploring the effectiveness of the associated impacts.

Secondly, I agree with your recommendations around the need for clarification around multiple aspects of the logic model and its implementation. A logic model can be fit for purpose but if it is misunderstood or incorrectly implemented it is of little practical use. There could be stronger language around how ill-defined key components of the programme appeared to be for participants within your data. For example, the discrepancy as to what the intended impact of the programme is constitutes a huge problem. As you highlight in the introduction the development outcomes need to be prioritised within SfD and a lack of clarity around them is a huge flaw within programming. I understand the need for sensitivity with partner organisations, but this is the purpose of this study and without such clarity future aims around effectiveness evaluation cannot occur. These recommendations exist within Table 2 but the importance/strength of these recommendations is not always mirrored in the discussion. These can be strengthened while still celebrating P3P for their willingness to engage with independent researchers to explore their intervention. 

Thanks also for this valuable comment. We added some explanations and a few sentences in the discussion section to address these points. See p.18, l.548-589.

 

REVIEWER 2

RESPONSES TO REVIEWERS

COMMENTS RESPONSES TO REVIEWER 2

This is a well written and organised paper, with a coherent narrative and argument, and which moves towards enhancing our knowledge of the use of logic models in the Sport for Development field. 

Thanks for your comments, there are very appreciated! You perfectly understand what is the aim of this study and what we intended to do. 

I have only a few suggested issues for the authors to consider:

- There could be a more rigorous discussion of the role of logic models in both the realm of sport (not just SfD) and the realm of social intervention (social work, community work, development work). SfD sits at the intersection of these two realms, so there needs to be some consideration of where logic models fit in, in these respective fields. 

We added a few sentences on this point. See p.19-20, l.548-589.

- Some justification of the focus on logic models compared to ‘theories of change’ should be given. The latter tend to be more prominent in SfD agencies and programs, so why focus on the former here? 

We added some elements to explain this point. See p.4 l.101-112 in the introduction.

- I’m surprised that there is not more discussion of Coalter’s work with respect to the critical evaluation of SfD programs. Coalter’s recent work on program theory may be particularly salient here. 

Thanks for your comment, we tried to rearrange this part following your proposition. We introduced his works (p.3, l.74-76, l.105) and we discussed Coalter’s work in the discussion section, quoting your recommendation (p.18, l.548-564). We consider your suggestion regarding Coalter’s recent work (Fred Coalter, Marc Theeboom & Jasper Truyens (2020) Developing a programme theory for sport and employability programmes for NEETs, International Journal of Sport Policy and Politics, 12:4, 679-697, DOI:10.1080/19406940.2020.1832136) on the program theory not perfectly fitting the content of our manuscript. We are already citing references quotes by Coalter (2020): Weiss 1995, 1998 and Pawson and Tilley 2004.

- The paper could and should respond more adequately in the conclusion to the routine ‘so what?’ question. There should be greater consideration of what these findings may mean for future research and studies, as well as practice, in the SfD field, and also, perhaps more importantly, for researchers in broader fields in sport or community/social development or youth work. These latter points are especially relevant given the audience of the journal, which is far wider than the sports studies field. 

Thanks for your comment, we tried to rearrange this part following your proposition. See p.19-20 et l.574-589.

---

## [Decision Letter · Decision Letter 1]

21 Mar 2022

PONE-D-21-03672R1Analyzing a SDP program’s logic model with key actors’ perceptions The case of Pour 3 Points organization in Montreal.PLOS ONE

Dear Dr. Gadais,

Thank you for submitting your manuscript to PLOS ONE. After careful consideration, we feel that it has merit but does not fully meet PLOS ONE’s publication criteria as it currently stands. Therefore, we invite you to submit a revised version of the manuscript that addresses the points raised during the review process.

Please consider all comments

We look forward to receiving your revised manuscript.

Kind regards,

Ahmed Mancy Mosa, Ph.D.

Academic Editor

PLOS ONE

Journal Requirements:

Reviewers' comments:

Reviewer's Responses to Questions

**Comments to the Author**

1. If the authors have adequately addressed your comments raised in a previous round of review and you feel that this manuscript is now acceptable for publication, you may indicate that here to bypass the “Comments to the Author” section, enter your conflict of interest statement in the “Confidential to Editor” section, and submit your "Accept" recommendation.

Reviewer #1: All comments have been addressed

2. Is the manuscript technically sound, and do the data support the conclusions?

Reviewer #1: Yes

3. Has the statistical analysis been performed appropriately and rigorously? 

Reviewer #1: N/A

4. Have the authors made all data underlying the findings in their manuscript fully available?

Reviewer #1: Yes

5. Is the manuscript presented in an intelligible fashion and written in standard English?

Reviewer #1: Yes

6. Review Comments to the Author

Reviewer #1: I thank the authors for their clear responses to previous comments. See below for remaining minor recommendations.

Language

Review language throughout, some long sentences could do with breaking up for clarity. Specific lines highlighted for review of sense/clarity:

Line 86 review use of term ‘real’ what is meant by this

Line 90-91 starting 'This literature review,' review for clarity

Line 108-110 starting 'Ridde and Dagenais,' review for clarity

Line 568-570 starting 'It could be considered,' review for clarity

Line 587-589 starting 'In reality, those,' review for clarity

Throughout paper remove use of 1st person (‘we’, ‘our’). Currently mixes 1st and 3rd person references to authors. Consistent use of 3rd person (such as ‘the authors’ or ‘the research team’) preferred.

Content

Line 221 clarify if interviews totalled 60mins or were on average 60mins each

Line 254, include your definition of bracketing as it has been utilised in varied manners across different forms of Qual research.

Line 259 onwards, good inclusion of positionality and background. A line about different contributors’ backgrounds professionally (relevant professional experiences and paradigms of interest) would further strengthen this section.

Language around impact much better but would suggest a final check on all language to ensure nothing could read as having been objectively measured. Example line 640 Findings reveal also that the program is successful for all key actors should be changed to Findings reveal also that the program is perceived as successful for all key actors. Such simple changes prevent any misinterpretation of impact findings.

Pending the above minor changes and a thorough spelling/grammar proof this paper should be ready for publication and prove a valuable addition to wider literature.

7. PLOS authors have the option to publish the peer review history of their article (what does this mean?). If published, this will include your full peer review and any attached files.

Reviewer #1: No

---

## [Author Response · Author response to Decision Letter 1]

2 Apr 2022

PONE-D-21-03672

Analyzing a SDP program’s logic model with key actors’ perceptions The case of Pour 3 Points organization in Montreal.

PLOS ONE

 

REVIEWER 1

COMMENTS RESPONSES TO REVIEWERS

Reviewer #1: I thank the authors for their clear responses to previous comments. See below for remaining minor recommendations. 

Thanks for your kind comments. 

Language

Review language throughout, some long sentences could do with breaking up for clarity. Specific lines highlighted for review of sense/clarity:

Line 86 review use of term ‘real’ what is meant by this

Line 90-91 starting 'This literature review,' review for clarity

Line 108-110 starting 'Ridde and Dagenais,' review for clarity

Line 568-570 starting 'It could be considered,' review for clarity

Line 587-589 starting 'In reality, those,' review for clarity

We made the changes 

l.78 we erased “real”

l.82 we erased “review”

l.100 we keep the sentences as it stands because we mentioned authors point of view.

l.558 we keep the sentences as it stands. This is exactly the idea we want to focus on.

l.577 we erased “in reality”

Throughout paper remove use of 1st person (‘we’, ‘our’). Currently mixes 1st and 3rd person references to authors. Consistent use of 3rd person (such as ‘the authors’ or ‘the research team’) preferred. 

We made all the changes

Content

Line 221 clarify if interviews totalled 60mins or were on average 60mins each 

We made the clarification

Line 254, include your definition of bracketing as it has been utilised in varied manners across different forms of Qual research. 

l. 247 We included our definition as you suggested

Line 259 onwards, good inclusion of positionality and background. A line about different contributors’ backgrounds professionally (relevant professional experiences and paradigms of interest) would further strengthen this section. 

L. 255 We added some elements regarding your comment.

Language around impact much better but would suggest a final check on all language to ensure nothing could read as having been objectively measured. Example line 640 Findings reveal also that the program is successful for all key actors should be changed to Findings reveal also that the program is perceived as successful for all key actors. Such simple changes prevent any misinterpretation of impact findings. 

Thanks for this relevant comment. We did a deep modification of those aspects inside the text.

---

## [Decision Letter · Decision Letter 2]

18 Apr 2022

Analyzing a Sport For Development program’s logic model with key actors’ perceptions The case of Pour 3 Points organization in Montreal.

PONE-D-21-03672R2

Dear Dr. Gadais,

We’re pleased to inform you that your manuscript has been judged scientifically suitable for publication and will be formally accepted for publication once it meets all outstanding technical requirements.

Kind regards,

Ahmed Mancy Mosa, Ph.D.

Academic Editor

PLOS ONE

Additional Editor Comments (optional):

Reviewers' comments:

Reviewer's Responses to Questions

**Comments to the Author**

1. If the authors have adequately addressed your comments raised in a previous round of review and you feel that this manuscript is now acceptable for publication, you may indicate that here to bypass the “Comments to the Author” section, enter your conflict of interest statement in the “Confidential to Editor” section, and submit your "Accept" recommendation.

Reviewer #1: All comments have been addressed

2. Is the manuscript technically sound, and do the data support the conclusions?

Reviewer #1: Yes

3. Has the statistical analysis been performed appropriately and rigorously? 

Reviewer #1: N/A

4. Have the authors made all data underlying the findings in their manuscript fully available?

Reviewer #1: Yes

5. Is the manuscript presented in an intelligible fashion and written in standard English?

Reviewer #1: Yes

6. Review Comments to the Author

Reviewer #1: (No Response)

7. PLOS authors have the option to publish the peer review history of their article (what does this mean?). If published, this will include your full peer review and any attached files.

Reviewer #1: No

---

## [Editor Report · Acceptance letter]

3 May 2022

PONE-D-21-03672R2 

Analyzing a Sport for Development program’s logic model by using key actors’ perceptions: The case of Pour 3 Points organization in Montreal. 

Dear Dr. Gadais:

I'm pleased to inform you that your manuscript has been deemed suitable for publication in PLOS ONE. Congratulations! Your manuscript is now with our production department. 

Kind regards, 

on behalf of

Dr. Ahmed Mancy Mosa 

Academic Editor

PLOS ONE